# The Severity of Pain and Comorbidities Significantly Impact the Dependency on Activities of Daily Living among Musculoskeletal Patients in Jizan, Saudi Arabia

**DOI:** 10.3390/healthcare11162313

**Published:** 2023-08-16

**Authors:** Mohammed A. Zalah, Hana I. Alsobayel, Fahad S. Algarni, Vishal Vennu, Zohoor H. Ajeebi, Hatem M. Maeshi, Saad M. Bindawas

**Affiliations:** 1Department of Rehabilitation Sciences, College of Applied Medical Sciences, King Saud University, Riyadh 11433, Saudi Arabia; 2Medical Rehabilitation Center, King Fahad Central Hospital, Jazan 82666, Saudi Arabia; 3Department of Physical Therapy, Ahad Al Mosaraha Hospital, Jazan 86289, Saudi Arabia; 4King Salman Center for Disability Research, Riyadh 11614, Saudi Arabia

**Keywords:** pain severity, comorbidity, physical function, older adults, musculoskeletal condition, Saudi Arabia

## Abstract

Limited research has been carried out on the effects of pain, comorbidity, and impaired function in musculoskeletal patients in Jizan, Saudi Arabia. A cross-sectional study was conducted on 115 patients (aged ≥ 55 years) with physician-diagnosed musculoskeletal conditions in Jizan to investigate the association between pain severity, comorbidities, and dependence on activities of daily living (ADLs). Self-reported questionnaires were used to collect data on pain, comorbidities, and physical function measured by ADLs. In ADLs, participants were categorized as dependent (n = 36) or independent (n = 79). Logistic regression analysis was employed to determine the predictors of dependence. The results showed that higher pain severity (adjusted odds ratio (OR): 1.69, 95% confidence interval (CI): 1.21–2.38, *p* = 0.002) and a greater number of comorbidities (adjusted OR: 1.52, 95% CI: 1.06–2.17, *p* = 0.021) were independently associated with dependence in ADLs. These associations remained significant even after controlling for covariates. This study concluded that patients with musculoskeletal conditions in Jizan who experience high levels of pain and comorbidities are at risk of dependence on basic daily activities. Therefore, addressing pain and comorbidities is crucial for maintaining independence and improving quality of life. Personalized rehabilitation programs are needed to manage these conditions in this region.

## 1. Introduction

Over five decades, the Kingdom of Saudi Arabia (KSA) has undergone an impressive expansion, gradually evolving into an epitome of civilization within its region. It has emerged as the most cultured nation in the Middle East through persistent growth and development, solidifying its position as a beacon of progress and refinement. Encompassing a vast area of 2.5 million square kilometers, Saudi Arabia is one of the largest countries in the entire Middle Eastern landscape [1]. In 2017, the estimated population approached 35 million, a testament to its vibrant and thriving community. Notably, the older adult demographic accounted for 5.6% of this population, highlighting the region’s commitment to ensuring the well-being and care of its senior citizens [2]. The percentage of older adults is expected to significantly increase, reaching 25% by 2050. This growth reflects the demographic shift toward an aging population. Additionally, life expectancy is predicted to rise from 74 to 82 years due to improvements in healthcare and quality of life [2]. Population aging in developing countries is rapidly occurring, while advanced countries are seeing more gradual aging after many decades of constant socioeconomic development [3]. Aging is a complicated process characterized by gradually deteriorating physiological function over time [4,5]. Age is the most critical risk factor for many neurodegenerative, metabolic, cardiac, and musculoskeletal (MSK) conditions [6].

According to the World Health Organization, MSK conditions are one of the significant causes of disability in the older population [7]. Saudi Arabia has a high prevalence of MSK conditions affecting older adults, the second-leading cause of disability [8]. The most important consideration for older adults with MSK conditions is their ability to independently perform the functional activities required for daily living [9]. Studies have shown that MSK conditions and advancing age are associated with decreased physical function (PF) in older adults, as deterioration of the MSK system function leads to impairments in performing activities of daily living (ADLs) [10]. Limitations in PF are significant due to their pervasive occurrence and associations with reduced quality of life (QoL), increased risk of disability, falls, depression, and increased healthcare costs [11,12]. Consequently, preserving PF has become a significant public health concern [13]. 

Understanding musculoskeletal aging and the risk factors associated with impaired PF is essential. Individual variations in the rate of decline in PF deteriorate with increasing age and MSK conditions [14], and several factors have been associated with PF decline in older adults with MSK conditions. MSK pain in older adults is a significant health concern owing to its high prevalence and detrimental effect on well-being [15]. Pain in older adults is linked to multiple adverse impacts on health, including physical and psychological deficits, physical inactivity, falls, depression, and dementia [16]. Additionally, comorbidity is common in older people and has significant implications for patients, families, and healthcare services [17]. Comorbidity is strongly correlated with restrictions in PF and mobility [17]. 

Prior observational research has established a significant association between pain and impaired PF, leading to increased disability in older adults and suggesting that pain might serve as a predictor for functional decline and disability progression [15]. In contrast, a longitudinal study reported that pain severity was not correlated with the onset of disability [18]. Comorbidity and disability have been closely associated in cross-sectional studies. This association has been confirmed in longitudinal studies showing an increased risk of loss of mobility and functional dependency due to several chronic diseases [19]. Limited research has been conducted on the influence of comorbidity on physical functioning in patients with MSK conditions. Although existing evidence suggests that pain and comorbidity have an impact on disability and functional independence, there is a dearth of studies exploring their effects on PF in older adults with MSK conditions, particularly in the Jizan region. This study aims to investigate the relationship between pain severity, multiple comorbidities, and PF impairment among patients with MSK conditions. 

## 2. Materials and Methods

### 2.1. Study Design and Participants

This study employed convenience sampling to recruit participants with MSK conditions. This study focused on individuals aged 55 years and older who were diagnosed with specific MSK conditions, including osteoarthritis (OA), low back pain, neck pain, shoulder pain, and plantar fasciitis. Orthopedic or physical medicine and rehabilitation physicians carried out the diagnosis of these MSK conditions. For inclusion in this study, participants had to meet the following criteria: (1) age 55 years or older, (2) diagnosis of one or more of the specified MSK conditions, and (3) confirmation of the diagnosis by qualified medical professionals. On the other hand, the following exclusion criteria were applied: (1) individuals with known mental disorders to minimize potential confounding factors, (2) those with severe disability related to the musculoskeletal system to ensure a more homogeneous study population, and (3) participants who had undergone surgical interventions related to the musculoskeletal system to focus on the impact of non-surgical management. The use of convenience sampling was considered appropriate as it allowed for efficient data collection from a targeted population within the outpatient physiotherapy department. This method facilitated the accessibility and practicality of recruiting participants, enabling this study to be conducted within the designated time frame.

G-Power program (version 3.1.9.6) was used to estimate the sample size, taking into account an alpha level of 0.05, a correlation coefficient of 0.3, and a power of 0.95. This study was approved by the institutional review board of King Saud University (E-21-5895) and the Jazan Health Ethics Committee (ethics number 2139), and all participants provided informed consent.

### 2.2. Exposure

Participants were asked to complete a self-reported questionnaire regarding demographic characteristics, including age, sex, height, weight, educational level, marital status, and employment status. The Numeric Pain Rating Scale (NPRS) assessed the pain severity. The NPRS is an 11-point scale ranging from 0 to 10, with 0 indicating no pain and 10 indicating extreme pain. The NPRS is a reliable and valid pain severity scale in older adults [20]. 

This study used a self-reported questionnaire that included information regarding their history of the following chronic diseases: diabetes mellitus; hypertension; OA; osteoporosis; cardiovascular disease; and gastrointestinal, endocrine, metabolic, chronic respiratory, neurological, and genitourinary disorders. Participants indicated if they had each comorbidity in a yes/no manner. We calculated each patient’s total number of chronic conditions based on their responses. This has been a feasible strategy used to collect medical data in many clinical and epidemiological studies [21].

### 2.3. Outcome

The basic ADL self-reported questionnaire was used to determine PF among older adults. The basic ADL questionnaire assessed the ability to independently carry out self-care activities, including feeding, dressing, personal hygiene, toileting, bathing, and ambulation [22]. The ADL instrument is valid and reliable for evaluating PF in older adults [23]. In this measure, the participants’ scores were from 0 to 6. These scores were transformed into binary scores of 0 or 1. Scores of 0 indicate independence in PF if participants could perform all six ADL without help. Scores of 1 demonstrated dependent on PF if participants could not accomplish one or more of the six ADLs [23].

### 2.4. Statistical Analysis

We conducted data analysis for this study using Statistical Package for the Social Sciences version 28.0 software program. Descriptive statistics were applied to all variables, where continuous variables were expressed as mean and standard deviation (SD), and categorical variables were presented as frequency and percentage. We employed several statistical tests to assess the relationship between participants’ covariates and physical function (PF). These included the chi-square test, Fisher’s exact test, and independent t-test to calculate *p*-values. Fisher’s exact test was used when more than 80% of the cells had an expected count of less than 5, replacing the chi-square test to ensure robustness in cases of small expected counts. We employed binary logistic regression models to explore the predictors of PF impairment, treating PF as the dependent variable and other relevant variables as independent variables. The odds ratio was used to represent the probability of PF impairment risk relative to the probability of no risk of PF impairment. A significance level of *p* < 0.05 was considered statistically significant at a 95% confidence interval (CI), providing a rigorous basis for interpreting the results. These statistical approaches provide comprehensive and reliable insights into the associations between pain severity, comorbidities, and physical function in the study population.

## 3. Results

### 3.1. Characteristics of the Study Participants

In this study, 115 participants over the age of 55 were analyzed, with 36 individuals (31.3%) classified as dependent and 79 individuals (68.7%) categorized as completely independent in ADLs (as shown in Figure 1). Table 1 provides a summary of the participants’ demographic characteristics. The sample consisted of 58 male participants (50.4%) and 57 female participants (49.6%), with an average age of 63.2 years (SD: 7.4). It is worth noting that the completely independent group had an average age that was five years younger than the dependent group. Most participants (79.1%) had completed intermediate school or below. Additionally, a significant number of participants were classified as overweight regardless of their dependency status. The mean NPRS score for the entire sample was 6.52 (SD = 1.9), with the dependent group exhibiting a higher mean NPRS score than the independent group, indicating more pain. Furthermore, the average number of comorbidities for the entire sample was 2.01 (SD = 1.6), with the dependent group demonstrating a significantly higher mean number of comorbidities than the independent group, indicating more medical conditions.

### 3.2. Association of Pain Severity and PF Status

The analysis in Table 2 highlights a strong connection between pain severity and PF (physical functioning) impairment in patients with musculoskeletal conditions. Model 1 demonstrates that a one-point increase in pain severity leads to a 54% increase in the likelihood of experiencing PF impairment, with an odds ratio (OR) of 1.54 (95% CI: 1.22–1.96, *p* < 0.001). Model 2, after adjusting for sociodemographic factors, still shows a significant association between pain severity and PF impairment, with an OR of 1.74 (95% CI: 1.25–2.42, *p* < 0.001). Even after considering age, sex, nationality, weight, height, educational level, marital status, and employment status, increasing pain severity independently leads to a 74% higher likelihood of PF impairment. Model 3 further confirms that pain severity is a significant predictor of PF impairment, with an OR of 1.69 (95% CI: 1.21–2.38, *p* = 0.002), even after accounting for comorbidities and sociodemographic variables.

### 3.3. Association of Comorbidities and PF Status

The data in Table 2 reveal a significant correlation between the number of comorbidities and physical function (PF) impairment. In Model 1, the analysis shows that each additional chronic disease increases the likelihood of PF impairment by 66% (OR = 1.66, 95% CI: 1.24–2.22, *p* < 0.001). Even after accounting for sociodemographic variables in Model 2, the link between comorbidities and PF impairment remains strong, with an OR of 1.59 (95% CI: 1.13–2.24, *p* < 0.001). This indicates that comorbidities independently contribute to a 59% higher chance of experiencing PF impairment, even after considering age, sex, nationality, weight, height, educational level, marital status, and employment status. Model 3, which incorporated pain severity, also demonstrated that the connection between comorbidities and PF impairment remained significant, with an OR of 1.52 (95% CI: 1.06–2.17, *p* = 0.021).

## 4. Discussion

In this study, the participants were all aged 55 years or older. Of these participants, 68.7% were found to be independent in terms of PF, while 31.3% were dependent. These findings align with those of previous research. For instance, a study carried out in Khamis Mushait, Saudi Arabia, found that 28.1% of participants experienced ADL impairment [24]. In an analysis conducted in Abha City, Saudi Arabia, 26.6% of individuals experienced declines in functional capacity during activities of daily living [25]. Age and educational levels were linked to ADL functional impairments in the study, consistent with previous research on aging and PF impairment [24,26]. Globally, lower levels of education have been linked to physical function impairment in older adults, according to studies conducted in India and southeastern Poland. Those with less education are at a higher risk for limitations in their daily activities [26,27].

In contrast, the covariates of sex and marital and employment status were not associated with impaired PF. In partial agreement with these results, a cross-sectional study found that sex was not significantly associated with ADL status in community-living adults aged 65 years or above [28]. Another study reported that marital status was not associated with a physical disability [29].

### 4.1. Association of Pain Severity and PF

The current study revealed a significant association between pain severity and impaired PF in older adults with MSK conditions. These findings are consistent with many previous studies that have suggested that pain severity may be an associated factor for adverse outcomes in persons with MSK conditions [30]. For example, a previous study found pain severity associated with impaired PF in Khamis Mushait’s older adults [24]. In a study conducted in Poland, higher pain severity increased the probability of PF impairment. There was a 27% significant increase in the risk of ADL impairment with each one-point increase on the pain scale [27]. A study showed that severe/extreme MSK pain intensity was significantly associated with disability, with an OR of 4.3 for severe/extreme pain and an OR of 1.54 for moderate pain among older adults [29]. A three-year longitudinal Japanese study found that MSK pain caused impaired PF and increased the risk of future disability (adjusted OR, 1.65) [31]. Moreover, in a study of 204 adults with MSK pain aged 50 years and older who evaluated their pain, the severity of pain was a significant predictor of disability across multiple aspects of daily living [32].

A study of 3097 Austrian seniors found that those with musculoskeletal conditions had more difficulty with daily activities. Pain severity was also found to predict impaired physical functioning [10]. Moreover, a cross-sectional study used 2017 Medical Expenditure Panel Survey data in the United States to examine characteristics associated with functional limitations among older adults aged 50 and older and found that pain intensity was associated with functional limitations. Those with increased pain intensity were likelier to report a functional limitation [33]. In addition, a cross-survey of the Spanish National Health Survey found that pain severity was the most crucial indicator of PF decline and caregiving demand in older adults with OA in Spain [34]. Furthermore, a double-blind randomized controlled trial assessed pain management by administering opioid analgesics to individuals with chronic back pain and discovered that acute opioid analgesic administration enhanced lifting capacity by 15% to 48%.

In contrast, reduced pain severity was associated with increased exercise performance, suggesting that pain severity significantly impacts functional ability [35]. However, we only examined pain severity and not pain location in the current research, making comparing these outcomes challenging. Therefore, pain severity has been significantly related to functional limitations without influence from the pain location, duration, and type [36]. Moreover, pain severity affects many of the older adult population. It is a crucial but under-recognized predictor of impaired PF in older adults with MSK conditions.

### 4.2. Association of Comorbidities and PF

The present study’s findings indicate a significant association between the number of comorbidities and impaired PF in older adults with MSK conditions. The findings of several studies support these results, noting a relationship between comorbidities and poor PF in older adults. For instance, the Health and Retirement Study in the United States found that multimorbidity was significantly associated with long-term PF impairments [17]. In addition, a cross-sectional study of women aged 80 years and older as part of the Women’s Health Initiative reported that comorbidities, quantified as a count of two to twelve chronic diseases that included overweight or obesity, were related to declining PF [37]. An analysis of older Mexican-American care recipients showed that three or more comorbidities were associated with more significant impairments in ADLs, particularly those with hypertension and OA plus diabetes, cognitive impairment, or CVD [38]. Furthermore, Scudds et al. (2000) [29] indicated that the number of chronic conditions was associated with a physical disability.

In a longitudinal study conducted in India, which utilized a nationally representative sample of individuals aged 60 and older, the importance of comorbidities and physical function (PF) was underscored. The study revealed that limitations in activities of daily living (ADLs) significantly increased in the presence of prior comorbidities [26]. Similarly, research conducted on a community-based sample of older adults in Shanghai, China, established a robust relationship between comorbidities and ADL impairments. Both studies highlight the crucial role comorbidities play in the daily functioning and well-being of older adults [39]. A longitudinal observational study conducted on older adults aged 60 or older discovered that a higher number of comorbidities, both at baseline and over time, led to a more substantial loss of physical independence [40]. In contrast, a prospective cohort study in Sweden demonstrated that individuals without any comorbidities maintained complete independence in their ADLs. These findings, along with the studies from India and China, emphasize the considerable impact of comorbidities on the daily functioning and well-being of older adults across different populations [19]. The severity of disability increased with a higher number of comorbidities, from 14.5% in subjects with one chronic disease to 17.0% in subjects with four or more conditions, emphasizing the strong association between the number of comorbidities and physical disability in old age. Similarly, the Survey of Health and Living Status of the Elderly in Taiwan found that increased chronic diseases led to functional impairments in older adults [41]. Older adults commonly have a higher number of comorbidities worldwide. As comorbidities increase with age, dependency in ADLs occurs [42]. Most studies have reported that functional disability increased along with the number of comorbidities in older adults, consistent with the current study’s findings. Impairment in PF is a symptom of deteriorating health due to the worsening of numerous chronic diseases in older adults.

### 4.3. Limitations and Strengths of This Study

This study has certain limitations that need to be acknowledged. Firstly, due to its cross-sectional design, it cannot establish causal relationships between variables. To verify the temporal nature of associations, longitudinal studies are required. Secondly, convenience sampling from a single physical therapy department may only represent some of the population of patients with musculoskeletal conditions in the region, limiting generalizability. Thirdly, self-reported pain severity, comorbidities, and physical function may be subject to recall bias and subjective interpretation, which may affect the accuracy of the findings. Objective assessments could have been incorporated to strengthen the findings. Additionally, confounding factors such as socioeconomic status, lifestyle, and access to care were not measured, although some efforts were made to account for them. This could affect the strength of observed associations. Fourthly, the small sample size of 115 participants provides limited statistical power to detect small effects. Therefore, a larger study is necessary to confirm these initial results. In the future, longitudinal studies with representative sampling, objective measures, consideration of additional confounding factors, and larger sample sizes across diverse populations can provide more conclusive evidence regarding associations between pain severity, comorbidities, and physical function impairment in patients with MSK conditions.

Although there are some limitations, this study showcases several strengths worth noting. Firstly, it sheds light on a topic that has yet to be extensively researched but is highly pertinent to the target population. This study addresses a significant knowledge gap by examining the relationship between pain, comorbidities, and physical function among patients with musculoskeletal conditions in Jizan, Saudi Arabia. Secondly, this study utilized well-established and validated measures and rigorous statistical analysis (binary logistic regression) to control potential confounding factors and explore relationships between key variables. This methodological rigor ensures the reliability and validity of the results. Thirdly, despite the relatively small sample size, this study included a diverse group of participants with various musculoskeletal conditions and pain levels, improving the generalizability of the findings to the patient population of interest. Lastly, this study grouped various musculoskeletal conditions together, which could limit the specificity of the findings due to potential differences in their causes and diagnostic criteria.

Overall, while this study is limited by sample size and the generalizability beyond the region, it does lay the groundwork for further research and has the potential for significant clinical impact. The findings provide valuable insights into this population’s contextually relevant management of musculoskeletal pain and comorbid conditions. In summary, the strengths of this study—novelty, clinical relevance, methodological rigor, generalizability to the target group, and potential for impact—outweigh the noted limitations. With further research, this study can potentially positively influence practice and outcomes.

## 5. Conclusions

This study highlights the critical impact of pain severity and the number of comorbidities on the ability of older adults with musculoskeletal conditions in Jizan, Saudi Arabia, to maintain independence. Our findings reveal a clear correlation between higher pain levels and a greater number of comorbidities dependent on daily activities in this demographic. These associations remained significant even after accounting for other factors, underscoring the importance of addressing these factors. Acknowledging the significance of severe pain and comorbidities in functional decline will aid in improving healthcare and overall health outcomes for older adults in Jizan. Focusing on personalized rehabilitation programs will empower this vulnerable population and promote healthy aging in the region.

## Figures and Tables

**Figure 1 healthcare-11-02313-f001:**
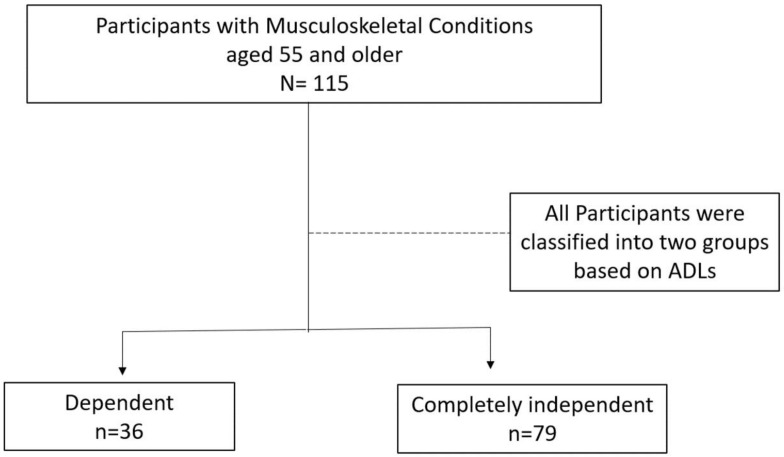
The flow of the study participants.

**Table 1 healthcare-11-02313-t001:** Demographic Characteristics Stratified by PF Status in the Study Participants.

Characteristics	Total Sample	Dependent	Completely Independent	*p*-Value
	n = 115	n = 36 (31.3%)	n = 79 (68.7%)	
**Age (years), mean ± SD**	63.2 ± 7.4	66.9 ± 9.5	61.2 ± 5.4	*<0.001*
≥65, n (%)	38 (33.0)	18 (50.0)	20 (25.3)	*0.009*
<65, n (%)	77 (67.0)	18 (50.0)	59 (74.7)	
**Sex, n (%)**				*0.386*
Men	58 (50.4)	16 (44.4)	42 (53.2)	
Women	57 (49.6)	20 (55.6)	37 (46.8)	
**Nationality, n (%)**				*1.000* *
Saudi	113 (98.3)	36 (100)	77 (97.5)	
Non-Saudi	2 (1.7)	0 (0.0)	2 (2.5)	
**Educational level, n (%)**				*0.001*
Intermediate school or less	91 (79.1)	35 (97.2)	56 (70.9)	
High school or more	24 (20.9)	1 (2.8)	23 (29.1)	
**Marital status, n (%)**				*0.565*
Married	99 (86.1)	30 (83.3)	69 (87.3)	
Not Married	16 (13.9)	6 (16.7)	10 (12.7)	
**Employment status, n (%)**				*0.431* *
Employed	7 (6.1)	1 (2.8)	6 (7.6)	
Not employed/ Retired	108 (93.9)	35 (97.2)	73 (92.4)	
BMI, mean ± SD	28.3 ± 4.6	29.0 ± 5.69	28.01 ± 3.97	*0.257*
NPRS, mean ± SD	6.57 ± 1.9	7.56 ± 1.9	6.11 ± 1.7	<*0.001*
No. of comorbidities, mean ± SD	2.01 ± 1.6	2.81 ± 1.6	1.65 ± 1.4	<*0.001*

Abbreviations: NPRS, numeric pain rating scale; BMI, body mass index. The independent *t*-test for continuous data was used to calculate the *p*-value. The chi-square test for categorical data was used to calculate the *p*-value. * Fisher’s exact test for categorical data was used to calculate the *p*-value.

**Table 2 healthcare-11-02313-t002:** Associations between Pain Severity, Comorbidities, and Physical Function: Binary Logistic Regression Analyses.

	Model 1			Model 2			Model 3		
	OR	95% CI	*p*-Value	OR	95% CI	*p*-Value	OR	95% CI	*p*-Value
Pain severity	1.54	1.22–1.96	*<0.001*	1.74	1.25–2.42	*<0.001*	1.69	1.21–2.38	*0.002*
No. of comorbidities	1.66	1.24–2.22	*<0.001*	1.59	1.13–2.24	*<0.001*	1.52	1.06–2.17	*0.021*

Abbreviations: OR, odds ratio; CI, confidence interval. Model 1 = unadjusted model. Model 2 = adjusted for sociodemographics (age, sex, nationality, weight, height, educational level, marital status, and employment status). Model 3 = adjusted for model 2 + pain severity (NPRS) variables and BMI.

## Data Availability

The data supporting this study’s findings are available on request from the corresponding author, S.M.B. The data are not publicly available due to restrictions, e.g., they contain information that can compromise the privacy of research participants.

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
