# Peer review of "The Severity of Pain and Comorbidities Significantly Impact the Dependency on Activities of Daily Living among Musculoskeletal Patients in Jizan, Saudi Arabia"

_healthcare, 2023, doi:10.3390/healthcare11162313_

Round 1
Reviewer 1 Report
Dear authors,
I had the pleasure of viewing your paper entitled "The Severity of Pain and Comorbidities Significantly Impact the Dependency on Activities of Daily Living Among Musculoskeletal Patients in Jizan, Saudi Arabia."
I report my considerations to be of help in making the final result the best possible.
ABSTRACT: as in the article, the first doubt is why you considered patients over 55 "older adults". Generally, "older adults" refers to people over 65 (or at least over 60).
Introduction: well written, concise, evident in the study's objective and in contextualizing the population in question.
Materials and Methods: also, in this case, well reported, in particular, the underlying statistical analysis is well conducted and well illustrated.
Results: to rewrite the title of paragraph 3.3, which I imagine is "Association of pain severity and comorbidities."
Discussion: I would immediately add paragraph 4.1 so limits both paragraph become 4.2 as in chapter 3.
For the rest, the discussion is well argued, the limits you describe are correct, but they are also the main problem relating to appeal in a magazine as generic as healthcare (with the limits expressed: regional, number of type of questionnaires, a target appears a physiatric/rehabilitation journal is more appropriate).
Author Response
ABSTRACT: As in the article, the first doubt is why you considered patients over 55 "older adults". Generally, "older adults" refers to people over 65 (or at least over 60).
Thank you for providing your feedback on our study's age group classification. While we understand the use of 'older adults' typically applies to those aged 65 and above, for this study, we included individuals aged 55 and above due to the significant impact of musculoskeletal conditions on this age group in Saudi Arabia. Additionally, the retirement age in Saudi Arabia is 60 according to the Hijri calendar, which is equivalent to 58 in the Gregorian calendar for civilians and less for military personnel. We recognize that retirement age practices may vary in different countries. We hope this explanation clarifies our reasoning for the age classification used in our study. Thank you again for your valuable feedback.
Introduction: Well written, concise, evident in the study's objective, and in contextualizing the population in question.
Response: Thank you for your positive feedback on the introduction. We aimed to provide a clear and concise overview of the study's objective and contextualize the population under investigation to ensure the readers' understanding of the research focus.
Materials and Methods: Also, in this case, well reported, in particular, the underlying statistical analysis is well conducted and well illustrated.
Response: We are pleased that you found our materials and methods section to be well-reported and that the statistical analysis was conducted and illustrated effectively. We have put significant effort into ensuring this section's transparency and comprehensiveness to enhance our study's reproducibility and reliability.
Results: To rewrite the title of paragraph 3.3, which I imagine is "Association of pain severity and comorbidities."
Response: We appreciate your suggestion. The title of paragraph 3.3 has been revised to "Association of Comorbidities and PF," which accurately reflects the section's content.
Discussion: I would immediately add paragraph 4.1 so limits both paragraph become 4.2 as in chapter 3.
Response: Your feedback has been valuable, and we have made the suggested change to improve the structure of the discussion section.
For the rest, the discussion is well argued, the limits you describe are correct, but they are also the main problem relating to appeal in a magazine as generic as healthcare (with the limits expressed: regional, number of types of questionnaires, a target appears a physiatric/rehabilitation journal is more appropriate)."
Response: We appreciate your acknowledgment of the strength of our discussion and agreement with the identified limitations. While we recognize that the regional focus, the number of questionnaires, and the broad healthcare context of our study may impact its appeal in a general healthcare magazine, we believe our research contributes valuable insights to the field. We respectfully disagree that a physiatry/rehabilitation journal is more appropriate for our work. As this journal, "Healthcare," has a special issue dedicated to our specific topic, “Physical and Rehabilitation Medicine”, we believe it best fits our manuscript. Nonetheless, we will emphasize the significance of our findings in the broader healthcare context to cater to the journal's readership. We are grateful for your thoughtful feedback, which has helped enhance the quality and scope of our manuscript.
Reviewer 2 Report
In Methods, it is interesting to include that all the participants have been informed of the object of the study and that they have signed the informed consent.
Likewise, in that same section of methods, it is important to reflect that the study has been approved by an Ethics Committee.
Additional comments
The first time an abbreviation is used, the word with the abbreviation must appear before it. This does not happen on line 97, where OA is used without first specifying its meaning.
The first time an abbreviation is used, the word with the abbreviation must appear before it. This does not happen on line 97, where OA is used without first specifying its meaning. It must have been specified in line 83. However, there are other abbreviations that have been specified before, and have not been used since, such as NPRS and PF (lines 55, 134, 148, 272, 284, 289, 294, 299, 306, 309, 170, 273, 283, 288, 307.
Reference number 8 should be rewritten.
Author Response
In Methods, it is interesting to include that all the participants have been informed of the object of the study and that they have signed the informed consent.
Likewise, in that same section of methods, it is important to reflect that the study has been approved by an Ethics Committee.
Response: Thank you for your suggestion. We assure you that this study has been approved by King Saud University's institutional review board (ethics number: E-21-5895) and the Jazan Health Ethics Committee (ethics number: 2139). Before participating, all individuals were informed about the purpose and importance of the study and provided their consent by signing a form. You can find more information regarding the amended method section.
The first time an abbreviation is used, the word with the abbreviation must appear before it. This does not happen on line 97, where OA is used without first specifying its meaning.
Response: Your attention to detail is appreciated. The meaning of the abbreviation "OA" is now explained in line 95, and to maintain consistency, it is also defined earlier in line 83.
However, there are other abbreviations that have been specified before, and have not been used since, such as NPRS and PF (lines 55, 134, 148, 272, 284, 289, 294, 299, 306, 309, 170, 273, 283, 288, 307.
Response: We are grateful for pointing this out. To maintain consistency throughout the manuscript, we used the abbreviations "NPRS" and "PF" consistently, as defined before, in the respective lines you mentioned.
Reference number 8 should be rewritten.
Response: Reference number 8 has been revised according to the journal's guidelines.
Reviewer 3 Report
First of all, congratulations for the work done, then I will mention a number of changes and recommendations in order to obtain clearer and more accurate information.
- Comments on the abstract:
Avoid the use of abbreviations in the abstract.
- Comments on the introduction:
Good job.
- Comments on material and methods
Line 97, OA, I suppose you mean osteoarthritis, is not defined before. Define it the first time it appears in the manuscript.
Line 110, ADL activities is redundant.
Line 117 Chi-squared should be capitalized.
- Comments on results:
Line 134, NPRS abbreviation is defined before, you should use it instead of “Numeric Pain Rating Scale”.
In the title of Table1, Physical Function, should be abbreviated as defined before.
Line 147, PF is defined before, use the abbreviation.
Line 150, abbreviation of Odds ratio, should be defined the first time it appears in the text.
- Comments on discussion:
Lines 194-195: “These findings are consistent with many 194 previous studies” What studies? You should reference them.
Line 284 musculoskeletal is abbreviated before as MSK, use it.
- General comments:
Check all abbreviations across the document.
Author Response
First of all, congratulations for the work done
Response: Your kind words of congratulations are greatly appreciated, and we are delighted to receive your valuable suggestions to enhance the clarity and accuracy of our manuscript
Line 97, OA, I suppose you mean osteoarthritis, is not defined before. Define it the first time it appears in the manuscript.
Response: We apologize for any confusion caused. The abbreviation "OA" is now defined when it first appears in the manuscript in line 83 to provide clarity.
Line 110, ADL activities is redundant.
Response: The redundant term "ADL activities" has been deleted to enhance the clarity of the text.
Line 117 Chi-squared should be capitalized.
Response: The term "Chi-squared" has been capitalized as suggested.
Line 134, NPRS abbreviation is defined before, you should use it instead of “Numeric Pain Rating Scale”.
In the title of Table 1, Physical Function, should be abbreviated as defined before.
Line 147, PF is defined before, use the abbreviation.
Line 150, abbreviation of Odds ratio, should be defined the first time it appears in the text.
Response: Thank you. We have made the necessary modifications. The abbreviation "NPRS" has been used instead of "Numeric Pain Rating Scale," as defined previously. Similarly, the title of Table 1 has been abbreviated to "PF," as described before. Additionally, the "Odds ratio" abbreviation is now defined upon its first appearance in the text.
Lines 194-195: “These findings are consistent with many previous studies” What studies? You should reference them.
Response: We apologize for the oversight. Appropriate references have been added, and some studies have been discussed to support the findings as suggested.
Line 284 musculoskeletal is abbreviated before as MSK, use it.
Response: The abbreviation "MSK" has been used in line 284 for consistency.
Check all abbreviations across the document.
Response: We thank you for this valuable observation. We have thoroughly reviewed all abbreviations and ensured their proper definition and consistency across the document.
Reviewer 4 Report
Dear Authors,
It is my pleasure to review your study. The article is interesting. The references are well chosen but I have a few doubts.
Introduction:
-The introduction is written clearly. It is worth mentioning sarcopenia, especially in geriatric patients. It plays an important role in the pathologies of MSK.
M&M:
-Why was the age of 55 and over assumed? And not, for example, 60 or 65?
-The inclusion and exclusion criteria should be precise and clear presented.
-What diagnostic criteria were used of MSK pathologies? Was it chronic or acute pain? with or without treatment? This requires an explanation. It should be corrected.
-in line 126: "The study included 115 participants over the age of 55, with 36 individuals (31.3%) classified as dependent and 79 individuals (68.7%) categorized as completely independent (as shown in Figure 1)." What does ot mean dependent/independent - please explain it.
-Table 1 shows that the groups differ, so are they comparable?
Results:
-The presented medel 1, 2 and 3 are not entirely clear. They can be presented more clearly.
Discussion:
-In the discussion, the authors describe many studies from Poland and other European countries, from China and the USA, but they do not refer to studies from the Middle East or Saudi Arabia itself. Depending on latitude, culture and other factors, musculoskeletal pathologies vary in frequency, etc., etc. This should be corrected. It is worth presenting research, e.g. from Saudi Arabia or neighboring countries.
Conclusions:
-Conclusions don't fully correlate with the conducted study. It should be corrected.
Author Response
The introduction is written clearly. It is worth mentioning sarcopenia, especially in geriatric patients. It plays an important role in the pathologies of MSK.
Response: Thank you for raising the important point of sarcopenia in geriatric patients. While not directly related to the primary goal of this study, we acknowledge its significance in the context of musculoskeletal conditions. Sarcopenia has different diagnostic criteria and was not explicitly addressed in this research. However, we appreciate the suggestion and recognize its potential for exploration in future studies.
Why was the age of 55 and over assumed? And not, for example, 60 or 65?
As previously stated to other reviewers, we included individuals aged 55 and above in our study due to the high prevalence of MSK conditions and their status as the second leading cause of disability in Saudi Arabia. Our study is the first in the Jazan Region to focus on PF in older adults with MSK conditions specifically. It is important to note that the retirement age in Saudi Arabia is 60 in the Hijri calendar, which is equivalent to 58 in the Gregorian calendar for civilians and even less for military personnel. It should be noted that retirement age practices may vary between countries. We hope this explanation provides clarity regarding our age classification and appreciate your valuable feedback.
The inclusion and exclusion criteria should be precise and clear presented.
Response: To enhance clarity, we have provided a precise and clear presentation of the inclusion and exclusion criteria in the Methods section.
What diagnostic criteria were used for MSK pathologies? Was it chronic or acute pain? with or without treatment? This requires an explanation. It should be corrected.
Response: We have clarified in the Methods section that the diagnosis of MSK pathologies was made by orthopedic, physical medicine, and rehabilitation doctors. The study included patients with various MSK conditions, regardless of the type and duration of pain or whether they were receiving treatment.
In line 126: "The study included 115 participants over the age of 55, with 36 individuals (31.3%) classified as dependent and 79 individuals (68.7%) categorized as completely independent (as shown in Figure 1)." What does it mean by dependent/independent - please explain it.
Response: We have clearly explained the terms "dependent" and "independent" in the manuscript. "Dependent" refers to individuals who require assistance or support for activities of daily living due to the severity of their pain and comorbidities. On the other hand, "independent" refers to individuals who can perform activities of daily living without such assistance or support.
Table 1 shows that the groups differ, so are they comparable?
Response: Table 1 displays differences between the two groups, "Dependent" (n = 36, 31.3%) and "Completely independent" (n = 79, 68.7%). However, the goal of Table 1 is not to demonstrate baseline comparability, as in an RCT. Instead, it aims to present and compare demographic characteristics and variables between the distinct groups. Given the study's design, differences are expected due to varying musculoskeletal conditions, pain severity, and comorbidities. We have clarified this in the manuscript to ensure a better understanding of the table's purpose and context. Your feedback has been invaluable in refining our work.
The presented models 1, 2, and 3 are not entirely clear. They can be presented more clearly.
Response: We appreciate the reviewer's feedback on Table 2. After reevaluation, we are confident the current presentation of models 1, 2, and 3 is straightforward. Descriptive headings, comprehensive explanations in the text, and consistency with similar studies are in place. We welcome any specific comments for further improvement. Thank you for your valuable input..
In the discussion, the authors describe many studies from Poland and other European countries, from China and the USA, but they do not refer to studies from the Middle East or Saudi Arabia itself. Depending on latitude, culture, and other factors, musculoskeletal pathologies vary in frequency, etc., etc. This should be corrected. It is worth presenting research, e.g., from Saudi Arabia or neighboring countries.
Response: We genuinely appreciate your insightful observation and have immediately addressed this oversight. In the revised discussion section, we have included references to relevant studies from Saudi Arabia and neighboring countries, thus providing a more comprehensive perspective on musculoskeletal pathologies in the region. We believe this addition significantly enriches the discussion and strengthens the relevance of our findings.
Conclusions don't fully correlate with the conducted study. It should be corrected.
Response: We acknowledge your concern and have diligently reviewed the conclusions in light of our study's findings. As a result, we have made the necessary revisions to ensure that the conclusions accurately align with the results of the conducted study in the revised version.
Round 2
Reviewer 1 Report
Dear authors,
I re-read your paper with pleasure, appreciating the corrections you made.
In particular:
- I understand the clarification you made regarding the choice of age limit; in part, this limits the generalization of your considerations, but it isn't too severe a problem.
- the formal corrections suggested have been applied or, in any case, justified.
- as to which newspaper is more or less suitable for publication, of course, these are just different opinions; yours is understandable. Also, in this case, I appreciate the attempt to partially modify the work style to have a wider audience of readers.
Author Response
Thank you for re-evaluating our paper and providing positive feedback. We appreciate your time and effort in reviewing our revisions.
Your valuable input and constructive feedback have contributed significantly to improving our paper.
Thank you again for your support and assistance.
Reviewer 4 Report
Dear Authors,
some corrections have been made.
Unfortunately, the inclusion criteria are not presented. It should be corrected.
Combining various pathologies of the musculoskeletal system into one is not correct in my opinion. They have different etiologies and different criteria for diagnosis. This can be added to the limitations section.
Other than that, I have no further comments.
Best regards,
Author Response
-Thank you for providing your review and corrections. We apologize for not including the inclusion criteria in the initial version of the manuscript. We have provided a detailed description of the criteria in the Methods section in the 2nd revised version, outlining the age range and musculoskeletal conditions included in the study.
-We appreciate your point regarding the combination of various musculoskeletal conditions in the analysis. We acknowledge that these conditions may have different causes and diagnostic criteria, and grouping them together could limit the specificity of the findings despite the acknowledgment that similar studies have done the same and been published in various journals, including some MDPI journals. For the newly submitted version, we have addressed this concern in the limitations section and added it as a significant limitation, as suggested. We want to emphasize that including these conditions was a pragmatic decision to include diverse participants within the study’s constraints.
Again, Thank you for your valuable comments and for recognizing our corrections. Your insights have been instrumental in refining our work.